# Understanding the Link between Sugar and Cancer: An Examination of the Preclinical and Clinical Evidence

**DOI:** 10.3390/cancers14246042

**Published:** 2022-12-08

**Authors:** Margeaux Epner, Peiying Yang, Richard W. Wagner, Lorenzo Cohen

**Affiliations:** 1Department of Anesthesiology and Pain Management, University of Texas Southwestern Medical Center, 5323 Harry Hines Blvd, Dallas, TX 75390, USA; 2Department of Palliative, Rehabilitation, and Integrative Medicine, The University of Texas MD Anderson Cancer Center, 1515 Holcombe Blvd, Houston, TX 77030, USA

**Keywords:** sucrose, fructose, cancer, inflammation, cancer metabolism, sugar

## Abstract

**Simple Summary:**

The average consumption of sugar in the US is significantly higher than the World Health Organization’s, the American Cancer Society’s, and the American Heart Association’s recommendations for daily sugar consumption. This review summarizes the research on the link between added sugar and cancer and the plausible mechanisms for a causal association. Evidence from epidemiologic and preclinical studies demonstrates that excess sugar consumption can lead to development of cancer and progression of disease for those with cancer independent of the association between sugar and obesity. The mechanistic preclinical studies in multiple cancers show that high-sucrose or high-fructose diets activate several mechanistic pathways, including inflammation, glucose, and lipid metabolic pathways.

**Abstract:**

Per capita sugar consumption has increased in the United States to over 45 kg per year. The average person in the US currently consumes significantly more added sugar in their diet than the World Health Organization’s, the American Cancer Society’s, and the American Heart Association’s recommendations for daily sugar consumption. Evidence from epidemiologic and preclinical studies demonstrates that excess sugar consumption can lead to development of cancer and progression of disease for those with cancer independent of the association between sugar and obesity. Human epidemiologic studies and mechanistic preclinical studies in multiple cancers support a causal link between excess sugar and cancer. Preclinical studies show that high-sucrose or high-fructose diets activate several mechanistic pathways, including inflammation, glucose, and lipid metabolic pathways. Although human studies are limited, compelling human and primate studies have explored the link between added sugar and metabolic syndrome (MetS), a risk factor for cancer. Substantial evidence suggests a causal link between MetS and added sugar, indicating important implications in the association between excess sugar consumption and cancer. Human clinical trials are needed to determine whether sugar increases cancer development and progression independently of its established role in causing obesity as well as for further exploration of the mechanisms involved.

## 1. Introduction

Historically, the highest cancer rates were in high-income countries, yet cancer is now ravaging low- and middle-income countries [1]. This is especially true for what are considered lifestyle- and obesity-related cancers, such as breast, prostate, colon and rectal, kidney, liver, pancreas, uterine, ovarian, and others. The “Westernization” of diets with increased consumption of highly processed foods and added sugars is viewed by many as the culprit [2]. The average United States resident consumes over 350 calories (approximately 21 tsp) of added sugar daily, which is significantly higher than the American Cancer Society’s, the World Health Organization’s (WHO), and the American Heart Association’s (AHA) recommendations for sugar consumption [3,4,5,6].

The Dietary Guidelines for Americans 2020–2025 [7], the ACS [4], and the WHO [8] recommend limiting added sugars to no more than 10% of total calories. For a 2000-calorie diet, that means about 200 calories from added sugars. The WHO has a further conditional recommendation to reduce added sugar consumption to less than 5% of calories, about 100 calories. This is more in line with the AHA recommendations of no more than 100 calories for women and 150 calories for men [6]. However, as can be seen in Figure 1, the US population consumption far exceeds even the more “generous” recommendations, consuming more than 350 calories from added sugars on average. Between the 1970s and the 1990s, there was a precipitous decrease in the consumption of refined cane and beet sugars, yet a sharp increase in high fructose corn syrup (HFCS). In the late 1990s, consumer and organizational pressures against using HFCS led to a reduction in HFCS consumption. Unfortunately, that reduction in HFCS consumption paralleled an increase in cane and beet sugars with the promise that being “natural” would be less harmful (Figure 1). Although total added sugar intake is down from 2000, total calorie intake increased from 2000 calories in 1970 to 2500 calories in 2010 (subsequent data not available). Furthermore, other calorie sources from high-glycemic-load, refined, fast-digesting carbohydrate foods (e.g., white flour, corn flower, and corn starch, etc.) have also increased, and, combined with added sugars, account for nearly 900 calories a day [5].

Sugary foods are perceived to be harmful primarily because they can cause weight gain. However, weight is not the only determinant of health: 20% of obese people have a normal metabolism, and 40% of people with normal body mass indices (BMIs) develop diabetes, hypertension, heart disease, and non-alcoholic fatty liver disease [9]. New research suggests that sugar plays a major role in the etiology of cancer and cancer progression [9]. Breast [10], colorectal [11], pancreatic [12,13], and other cancers [14,15] may be linked to added sugar, and in most cases independent of obesity and weight gain. Preclinical studies using mouse models show increased tumor burden [16], earlier onset [17], and a greater prevalence [18] of various cancers in mice fed high-sucrose or high-fructose diets compared with isocaloric starch diets. While no human clinical trials have explored the relationships between sucrose, fructose, and cancer, epidemiological studies and preclinical research in mice have found a strong association between excess sugar consumption and cancer. In addition, the literature from primate and human studies in the field of metabolic syndrome (MetS) indicates a need for further sugar-related research [19].

The role of added sugar in cancer development and progression is controversial. This paper aims to: (1) provide an overview on the evolution of sugar consumption and cancer incidence worldwide; (2) review current research linking dietary added sugars and cancer risk, prevalence, progression, and illness burden; and (3) explore plausible mechanisms linking added sugars to cancer.

### 1.1. Sugar and Cancer: The Past and Present

#### 1.1.1. What, Exactly, Are We Eating?

The per capita consumption of processed sugar in the US has surged to over 45 kg/year [20]. The increased consumption of added sugars, particularly sugar-sweetened beverages (SSB), is a pivotal contributor to worldwide epidemics of obesity, diabetes, heart disease, and cancer. Now that sodas in the US are sweetened mainly with HFCS, consumption of HFCS has increased more than 1000% between 1970 and 1990 [21], representing the greatest change in consumption of any food or food group in the United States. HFCS currently represents more than 40% of caloric sweeteners added to foods and beverages in the US [21], with some consuming as many as 316 daily kcal from HFCS alone, contributing to the worldwide obesity epidemic [21]. 

Sucrose, or table sugar, is a disaccharide comprising equal parts fructose and glucose. Glucose, a monosaccharide, is found in all carbohydrates and starchy foods and serves as the key source of energy for all animals, including humans, during cellular metabolism. Fructose, also a monosaccharide, is naturally found in fruit but has since been harnessed as a prominent added caloric sweetener in many foods.

Fructose is digested, absorbed, and metabolized differently from glucose. Unlike with glucose, hepatic metabolism of fructose favors de novo lipogenesis [21]. In addition, fructose found in HFCS, sugar, and certain foods does not stimulate production of insulin or leptin, both of which regulate food intake by increasing satiety and inhibiting hunger [18]. Thus, excess fructose in food likely contributes substantially to the current weight gain and obesity epidemic worldwide [21]. Therefore, added sugars containing fructose must be studied further to determine whether their impact on chronic diseases, especially cancer, is due mainly to their propensity to cause obesity or to a separate, more specific mechanism.

#### 1.1.2. The Westernization of Diet and Cancer Rates

Cancer was previously considered a disease of the affluent. However, low- and middle-income countries (LMIC) now make up 57% of cancer cases worldwide. The World Health Organization (WHO) estimates that, in only two decades, the rates will reach 22 million new cancer cases and 13 million cancer deaths annually [1], an estimated 57% increase in annual new cancer cases and a 65% increase in annual cancer deaths [22]. Countries such as Brazil, India, and China, which previously reported low rates of breast, prostate, and colon cancer, are now seeing significant increases in the incidence and mortality of these cancers [23,24]. A case-control study examining breast cancer incidence among women of Chinese, Japanese, and Filipino ethnicity in California and Hawaii found that those born in the United States had a 60% higher risk of breast cancer than those born in Asia [23]. Furthermore, Asian American women born in the United States with all four grandparents born in Asia had incidence rates similar to those of white women living in the same geographic area [23]. Breast cancer has now surpassed cervical cancer as the leading cause of cancer death among women in LMICs [22]. In Brazil, breast cancer mortality rates are also increasing steadily, with the highest average rates in the more urban southern and southeastern regions where São Paulo and Rio de Janeiro are located [25]. In 1998, prostate cancer accounted for 32% of cancer cases in men in the US but for less than 1% of all male cancers in men in Shanghai [26]. Prostate cancer mortality rates in the US have steadily declined by more than 40% between 1999 and 2017 [27]. In contrast, prostate cancer mortality rates in China increased by 5.5% annually between 2000 and 2011 [27]. In India, cancer incidence increased 1.1 to 2.0 percent per year between 2010 and 2019 [28], with breast cancer being the most common cancer in women and lung cancer for men [24]. India and China also have the highest incidence and number of people living with diabetes [29], a known risk factor for many cancers [30]. By 2040, global cancer cases will increase by over 40%, and it is estimated that two-thirds will occur in LMICs [31]. Changing diets, including consumption of fast-foods, highly processed foods, and excess sugar consumption, are hypothesized as a causative factor in the increasing incidence of cancer in LMICs.

According to food consumption trends in 2010, diets worldwide are undergoing Westernization, becoming more energy-dense and sugary than ever before [2]. LMICs, especially in Asia, Latin America, and Africa, have seen astonishing spikes in sugar consumption in recent years [2]. In 2002, the average Brazilian consumed 50.2 g of sugar per day, a number that has continued to grow each year [32]. The annual increase in sugar consumption in China between 2000 to 2007 was 2.2%, and is expected to double to 4.4% annually over the next few decades [33]. In addition, inhabitants of these LMICs received 54% to 70% of their daily calories in cereals alone [2]. The ubiquity of American fast- and processed foods and the overall Westernization of diets around the world has been hypothesized as the cause for the increased incidence of non-communicable diseases [34,35]. 

#### 1.1.3. Changing Perceptions and Guidelines about Sugar

Research on the relationship of added sugar and non-communicable diseases dates back over 50 years. During the 1960s and 1970s, physiologist John Yudkin identified sugar as a key cause of coronary heart disease (CHD) [36,37]. Fearing the impact of such results on the sugar industry, the Sugar Research Foundation paid two scientists at the Harvard University School of Public Health Nutrition Department to write a literature review, later published in the *New England Journal of Medicine* in 1967 [36]. The review questioned the validity of any study in which the research implicated sucrose in worsening CHD and instead blamed food high in saturated fats and cholesterol [36]. 

The resulting *1980 Dietary Guidelines for Americans* recommended lowering saturated fat and cholesterol intake to prevent CHD [36]. With these guidelines came an era of low-fat diets and low-fat/fat-free processed foods. Soon, much of the fat in processed food was replaced with sugar, and sugar became nearly unavoidable in the American diet. The sugar industry continues to fund research on CHD and other chronic diseases, indirectly influencing decades of American policy and health [36]. In 2003, when the WHO halved its sugar intake recommendation, the US Sugar Association pressured the US government to cut funding for the WHO if the recommendations were not changed [38]. While the AHA has since changed its recommendations to reflect the current knowledge that sucrose directly causes heart disease [39], other chronic disease institutions are lagging behind [40]. The current AHA guidelines recommend a daily limit of six teaspoons (30 g, or 120 calories) of added sugar for women and nine teaspoons (45 g, or 180 calories) for men [6]. The WHO also recommends dietary sugar intake less than 10% of daily energy intake (50 g per 2000 daily calories) and conditionally recommends that less than 5% of daily energy intake consist of added sugar. 

Despite studies showing the potential harms of added sugar and the important etiologic role it plays for many diseases, the current cancer dietary guidelines do not reflect this knowledge [9,41,42,43,44]. The American Institute for Cancer Research states, “There is no strong evidence that directly links sugar to increased cancer risk” [41] and recommends generally reducing sugar intake to avoid weight gain, but no specific guidelines are provided [41]. Additionally, none of the leading institutions in cancer research have substantial educational material or dietary guidelines on their websites regarding dietary sucrose. They either omit any mention of sugar completely or state that sugar and cancer may be linked only indirectly through weight gain [42,43,45].

The WHO issued a press release in February 2014 calling for quick and effective cancer prevention measures, entailing adequate legislation, taxation, and regulation of various carcinogenic agents, including SSB [1]. Additionally, the WHO strongly recommended dietary sugar intake less than 10% of daily energy intake (50 g per 2000 daily calories) and conditionally recommended that less than 5% of daily energy intake consist of added sugar [3]. Dietary cancer guidelines and federal and state policies also need to incorporate the knowledge that added sugar can be directly harmful.

## 2. Materials and Methods

### Search Strategy and Selection Criteria

References for this review were identified through searches of Ovid MEDLINE and PubMed with the search terms “sugar”, “sucrose”, “fructose”, “sweets”, “dessert”, “cancer”, “tumor”, “neoplasm”, carcinogenesis”, “breast neoplasm”, “neoplastic processes”, “neoplasm metastases”, “arachidonate 12 lipoxygenase”, “peroxisome proliferator-activated receptor”, “metabolic syndrome”, “insulin like growth factor 1”, “inflammation”, “immune system” from 1946 until present. Articles were also identified through searches of the authors’ own files and examining references sections of each article selected for review. Only papers published in English were reviewed. The final reference list was generated based on originality and relevance to the broad scope of this review. 

## 3. Results

### 3.1. Epidemiologic Studies Linking Sugar to Cancer

The following sections review epidemiological studies examining the association between added sugars and cancer risk and/or mortality. In the tables where the “Main Findings” are reported, we present the outcomes from the final models, controlling for multiple covariates. Importantly, 22 of 24 studies controlled for BMI (in some cases before and after diagnosis) and various other factors associated with cancer (e.g., smoking history, age, physical activity, and other dietary factors). The final statistical model of most studies described below found an association between sugar consumption and cancer outcomes independent of these other factors, suggesting unique risks associated with excess sugar consumption independent of other lifestyle factors, including BMI. 

#### 3.1.1. Breast Cancer

Numerous epidemiologic studies have shown an association between sugar and breast cancer (Table 1) [46,47]. Additionally, sucrose intake during adolescence [48] was significantly correlated with higher percentage of dense breast volume [49], a known risk factor for breast cancer [50].

In a case-control study in the United States, women under age 45 who consumed sweets 9.8 times per week or more experienced significantly higher breast cancer risk than those who consumed sweets less than 2.8 times per week [51]. The study found no significant association between risk of breast cancer and calorie intake, macronutrients, or types of fat, showing a sugar-specific association [51]. Similarly, a case-control study conducted in Italy found that women with the highest intake of desserts and sugars had multivariate odds ratios (OR)s of 1.19 (95% confidence interval (CI) 1.02–1.39) and 1.19 (95% CI 1.02–1.38), respectively, for breast cancer [47]. A French study found that sugary drinks were significantly associated with increased risk of breast cancer, with a hazard ratio (HR) of 1.22 (95% CI 1.07–1.39) [52]. 

While most research in this field has been conducted in high-income countries, one case-control study in Malaysia also found a significant two-fold increase in breast cancer risk with high sugar intake among both premenopausal (OR = 1.93, 95% CI 1.53–2.61) and postmenopausal participants (OR = 1.87, 95% CI 1.03–2.61) [10]. Taken together, findings in high- and LMICs show a consistent association between sugar consumption and increased risk of breast cancer. 

Sugar intake is also associated with increased risk of cancer-specific and all-cause mortality after a diagnosis of breast cancer. Consuming sugar-sweetened soda ≥5 times weekly vs. never/rarely was associated with total (HR = 1.62; 95% CI, 1.16–2.26; P_trend_ < 0.01) breast cancer mortality (HR = 1.85; 95% CI, 1.16–2.94; P_trend_ < 0.01) among women diagnosed with invasive breast cancer [53]. Similarly, Farvid at al. [46] examined 8863 women with stage I to III breast cancer who were part of the Nurses’ Health Study and found that women who had SSB consumption after diagnosis greater than zero to one serving per week had higher breast-cancer-specific mortality (>1 to 3 servings per week: HR = 1.31 [95% CI, 1.09–1.58]; >3 servings per week: HR = 1.35 [95% CI, 1.12–1.62]; P_trend_ = 0.001) and all-cause mortality (>1 to 3 servings per week: HR = 1.21 [95% CI, 1.07–1.37]; >3 servings per week: HR = 1.28 [95% CI, 1.13–1.45]; P_trend_ = 0.0001). In addition, replacing SSBs with coffee (18%) or tea (15%) reduced breast-cancer-specific mortality, and coffee (19%), tea (17%), or water (9%) lowered all-cause mortality risk [46].

**Table 1 cancers-14-06042-t001:** Added sugar intake and risk of developing breast cancer and mortality.

Author	Study Population	Study Design	Measure	Associations Examined	Main Findings *
Potischman et al. (2002) [51]	568 women with breast cancer, 1451 women without breast cancer	Cross-sectional	Food frequency questionnaire	Association between dietary patterns and breast cancer risk	Consuming sweets 9.8 times vs. <2.8 times per week increased risk of early-stage breast cancer in a linear manner (OR = 1.32, 95% CI 1.0–1.8)
Tavani et al. (2006) [47]	2569 women with breast cancer, 2588 without breast cancer	Cross-sectional	Interviewer-administered food frequency questionnaire adapted for Italy	Association between sugar intake and breast cancer risk	Risk of breast cancer was increased from lowest tertile to highest tertile for consumption of desserts (OR = 1.19, 95% CI 1.02–1.38) and for total sugars (OR = 1.19, CI 1.02–1.39)
Sulaiman et al. (2014) [10]	382 women with breast cancer, 382 without breast cancer	Population-based case-control	Food frequency questionnaire adapted for Malaysian population	Association between carbohydrate, fiber, and sugar intake and risk of breast cancer	Sugar intake was associated with increased risk of breast cancer in premenopausal (OR = 1.93, 95% CI: 1.53–2.61) and postmenopausal women (OR = 1.87, 95% CI: 1.03–2.61);
Jung et al. (2018) [49]	182 young women assessed at 10–19 years old and follow-up at 25–29 years old	Single arm, longitudinal	3 × 24 h dietary recalls at 2 weeks and 1, 3, and 5 years; glycemic index and glycemic levels; and breast density	Association between sucrose intake, fructose intake, and percentage of dense breast volume	Mean dense breast volume in first and fourth quartiles: 16.6% vs. 23.5% for sucrose and 17.2% vs. 22.3% for premenarcheal total carbohydrates, all P_trend_ ≤ 0.02
Farvid et al. (2021) [46]	8863 women with stage I–III breast cancer	Prospective cohort study	Food frequency questionnaire	Association between SSB/ASB use after diagnosis and cancer/all-cause mortality	Compared to zero, breast-cancer-specific mortality was associated with: 1< to 3 SSB drinks/week (HR = 1.31; 95% CI = 1.09–1.58); 3< SSB drinks/week (HR = 1.35; 95% CI = 1.12 –1.62). All-cause mortality: 1< to 3 SSB drinks/week (HR = 1.21; 95%CI = 1.07–1.37); 3< SSB drinks/week (HR = 1.28; 95%CI = 1.13–1.45). ASB not linked to breast cancer/all-cause mortality. One ASB for SSB drink/day replacement did not reduce risk, but replacement of SSB with coffee (18%, 15%) and tea (19%, 17%) reduced breast-cancer-specific and all-cause mortality risk, respectively.
Koyratty et al. (2021) [53]	927 women with invasive breast cancer followed for 18.7 years	Prospective cohort study	Food frequency questionnaire recall of sugar-sweetened soda consumption	Association between sugar-sweetened soda consumption weekly with overall and breast cancer mortality	≥5 times weekly vs. none/rarely increased overall mortality (HR = 1.62; 95% CI, 1.16–2.26; P_trend_ < 0.01) and breast cancer mortality (HR = 1.85; 95% CI, 1.16–2.94; P_trend_ < 0.01)

* Outcomes reported are from the final regression models that controlled for body mass index, as well as other factors associated with breast cancer. ASB = artificially sweetened beverages; CI = confidence interval; HR = hazard ratio; OR = odds ratio; SSB = sugar-sweetened beverages.

#### 3.1.2. Colorectal Cancer

Sugar may also play a role in the development and progression of colon cancer (Table 2). In a prospective cohort study of colon cancer patients, consuming two or more servings of SSB daily significantly increased risk of recurrence by 75% and risk of mortality compared to those who consumed less than two servings of SSB daily (95% CI 1.04–2.68) [11]. When further adjusted for dietary glycemic load in the multivariate model, the results remained nearly unchanged, suggesting a strong role for sugar [11]. 

One case-control study found total sucrose intake positively associated with a more than two-fold increase in risk of colorectal cancer and a significant dose–response gradient (OR 2.18, 95% CI 1.35–3.51) [54]. Contrary results in a pooled analysis of prospective cohort studies found no significant increase in colon cancer risk due to sugar-sweetened carbonated beverage intake (95% CI 0.66–1.32) [55], suggesting that more research is needed to better understand the role of added sugar in colon cancer development.

**Table 2 cancers-14-06042-t002:** Added sugar intake and risk of developing colorectal cancer and mortality.

Author	Study Population	Study Design	Measure	Associations Examined	Main Findings *
De Stefani et al. (1998) [54]	289 with colon cancer, 564 without colon cancer	Case-control study	Food frequency questionnaire	Association between sucrose and glucose and risk of colon cancer	Highest vs. lowest quartile of sucrose intake showed increased risk of colon cancer (OR = 2.18, 95% CI 1.35–3.51); did not control for BMI
Zhang et al. (2010) [55]	731,441 adults with colon cancer	Prospective cohort study	Food frequency questionnaire	Association between coffee, tea, and SSB and colon cancer risk	Non-significant colon cancer risk from SSB intake (OR = 0.94, 95% CI 0.66–1.32)
Fuchs et al. (2014) [11]	1011 colon cancer patients	Single cohort	Food frequency questionnaire	Association between SSB consumption and cancer recurrence or mortality	≥2 SSB per day vs. <2 SSB per month increased recurrence (HR = 1.75, CI 1.01–1.46) and mortality (HR = 1.62, CI 1.02–1.44); furthermore, recurrence/mortality was exacerbated for those ≥2 SSB per day and who were both overweight and less active (HR = 2.22, 95% CI 1.29–3.81)

* Outcomes reported are from the final regression models that controlled for body mass index (except where indicated), as well as other factors associated with colorectal cancer. BMI = body mass index; CI = confidence interval; HR = hazard ratio; OR = odds ratio; SSB = sugar-sweetened beverages.

#### 3.1.3. Pancreatic Cancer

A strong body of evidence suggests that a sucrose- and/or fructose-filled diet is associated with increased risk of pancreatic cancer, but other studies reported a weak association between added sugar intake and risk of pancreatic cancer (Table 3). A systematic review and meta-analysis of 10 cohort studies [56] found significant associations between fructose consumption and pancreatic cancer risk (relative risk (RR) = 1.22, 95% CI 1.08–1.37) [56]. One study showed a non-significant 53% increase in pancreatic cancer with high carbohydrate and sucrose intake [12], and, more specifically, found that high glycemic load and fructose intake were strongly associated with pancreatic cancer in overweight women [12]. Additionally, a prospective study demonstrated that higher consumption of sugar, soft drinks, and sweetened fruit soups or stewed fruit was associated with significant increases in pancreatic cancer risk of 69%, 93%, and 51%, respectively [57]. A multiethnic cohort study in Hawaii and Los Angeles documented a similar association between high fructose intake and pancreatic cancer [13]. In contrast, one study found that juice and soft drink consumption was not associated with risk of pancreatic cancer, and another study found that juice or nectar consumption was associated with a decrease in pancreatic cancer risk [58,59]. However, the authors suggest the results should be interpreted with caution as juices and nectars are usually rich in added sugars and fructose, which could potentially increase pancreatic cancer risk. It is important to note that, in all these studies, they controlled for BMI. More research is needed to improve understanding of the role of added sugar in the risk of pancreatic cancer.

#### 3.1.4. Miscellaneous Cancers

Studies of other cancer types also find sugar intake as a risk factor for cancer (Table 4). All but one study [15] controlled for weight and/or BMI, suggesting that the associations were independent of the harms of weight and weight gain. A large longitudinal and observational study found that daily consumption of only 100 mL of sugary drinks, including fruit juices, significantly increases the risk of overall cancer by 18% [52]. One review of 15 epidemiologic studies examining sugar intake and cancer [60] found positive associations between added simple-sugar and pancreatic, prostate, and liver cancer; hepatocellular carcinoma, lymphoma, and leukemia; cancer of the colon, breast, and small intestine; and cancer in general [60]. In a large prospective study of 435,674 participants, added sugars were significantly associated with an increased risk of esophageal adenocarcinoma, added fructose was significantly associated with a greater risk of small intestine cancer, and all sugars (total, sucrose, fructose, added sugars) were associated with an increased risk of pleural cancer [14]. Conversely, all the sugars were inversely correlated with ovarian cancer risk in women, and no association was found between any dietary sugars and risk of any other major cancer [14].

The Framingham Offspring Cohort (1991–2013) prospective study analyzed dietary-questionnaire data and cancer incidence and found no significant associations between sucrose, fructose, sugary foods, or sugary beverages with any site-specific cancers [61]. However, a 58% increased risk of prostate cancer was associated with higher consumption of fruit juices (>7 servings/week) [61]. Additionally, Jackson et al. found that a diet high in carbohydrates, including SSB, was positively associated with increased risk of prostate cancer [62]. Another case-control study found that sucrose consumption was positively associated with an increased risk of lung cancer [15]. A 70,000-person prospective study found that both men and women experienced significantly increased risk for extrahepatic biliary tract cancer and gallbladder cancer with high consumption of sugar-sweetened and artificially sweetened beverages [63]. Stepien et al. found that people who consumed more than six soft drinks per day had a significantly increased risk of developing hepatocellular carcinoma compared with non-consumers (HR = 1.83, 95% CI 1.11–3.02) [64]. Finally, a population-based case-control study found that excess sugar consumption was associated with shorter survival time among patients with esophageal cancer (HR for fourth vs. first quartile: 1.88; 95% CI 1.29–2.72) [65].

More recently, McCullough et al. [66] reported that, in a cohort of almost 1 million individuals with consumption of ≥2 SSB drinks/day vs. never, SSBs were associated with increased mortality from colorectal (HR = 1.09; 95% CI, 1.02–1.17; P_trend_ = 0.011) and kidney (HR = 1.17; 95% CI, 1.03–1.34; P_trend_ = 0.056) cancers, even after controlling for BMI. SSB consumption was also associated with mortality from obesity-related cancers, but the effect disappeared when controlling for obesity.

**Table 4 cancers-14-06042-t004:** Added sugar intake and risk of developing cancers and mortality (mixed cancers).

Author	Study Population	Study Design	Measure	Associations Examined	Main Findings *
De Stefani et al. (1998) [15]	463 with lung cancer, 465 controls	Case-control study	64-item food frequency questionnaire	Association between dietary patterns and lung cancer risk	Undifferentiated small-cell lung cancer risk was significantly associated with sucrose-rich food intake (OR = 3.7, 95% CI 1.4–10.0), sucrose-to-fiber ratio (OR = 2.3, 95% CI 1.2–4.5), and glycemic index (OR = 9.7, 95% CI 3.2–29.7); significant associations found with undifferentiated large cell lung cancer for sucrose-rich food intake (OR = 4.8, 95% CI 1.2–19.0) and glycemic index (OR = 13.6, 95% CI 1.7–109.0); no significant associations were found for squamous cell and adenocarcinoma cancers. The final models did not control for BMI.
Tasevska et al. (2012) [14]	435,674 adults with multiple cancers	Prospective cohort study	Food frequency questionnaire	Association between total sugars, fructose, sucrose, added fructose, added sucrose, and added sugars, and risk of 24 malignancies	Highest vs. lowest quintile of added sugar showed a positive association with risk for esophageal adenocarcinoma (HR = 1.62, 95% CI 1.07–2.45) and pleural cancer (HR = 2.20, 95% CI 1.16–4.16)
Jackson et al. (2013) [62]	243 with prostate cancer, 273 controls	Case-control study	124-item food frequency questionnaire	Association between dietary patterns and low- vs. high-grade prostate cancer	Highest vs. lowest tertile of refined carbohydrate intake was associated with greater total prostate cancer risk (OR = 2.02, 95% CI 1.05–3.87) and greater risk of low-grade disease (OR = 2.91; 95% CI 1.18–7.13)
Larsson et al. (2016) [63]	70,832 adults with biliary tract cancer	Prospective cohort study	Food frequency questionnaire	Association of SSB with risk of biliary tract cancer	≥2 vs. 0 sweetened beverages per day significantly increased risk of biliary tract (HR = 1.79, 95% CI 1.02–3.13), and gallbladder cancer (HR = 2.24, 95% CI 1.02–4.89)
Stepien et al. (2016) [64]	477,206 adults with hepatocellular carcinoma (HCC), intrahepatic bile duct (IHBC), and biliary tract cancers (GBTC)	Prospective cohort study	Country-specific dietary questionnaire	Association between SSB, sweetened drinks, and fruit and vegetable juices with risk of HCC, IHBC, and GBTC	>6 servings of combined SSB per week was positively associated with HCC (HR = 1.83, 95% CI 1.11–3.02); <1 serving of juice per week was associated with decreased HCC risk (HR = 0.60, 95% CI 0.38–0.95)
Miles et al. (2016) [65]	601 patients with upper aerodigestive tract (UADT) cancer	Population-based case-control study	Food frequency questionnaire	Association between SSB and UADT cancer	Highest vs. lowest quartile of SSB was associated with increased risk of UADT (HR = 1.88, 95% CI 1.29–2.72), and servings of sugary beverages (HR = 1.97, 95% CI 1.32–2.93) showed poorer survival in UADT
Makarem et al. (2018) [61]	3184 adults with breast, prostate, and colon cancers	Prospective cohort study	Food frequency questionnaire	Association between dietary sugars and sugary foods and adiposity-related cancers	>7 fruit juice servings per week (HR = 1.58, 95% CI 1.04–2.41) was associated with increased risk of prostate cancer
Chazelas et al. (2019) [52]	101,257 adults with multiple cancers	Prospective cohort study	Web-based 24 h dietary record	Association between sugary beverage consumption and risk of breast, colorectal, and prostate cancers	Positive associations were found between SSB and overall cancer (HR = 1.18, 95% CI 1.10–1.27) and breast cancer (HR = 1.22, 95% CI 1.07–1.39); 100% fruit juice consumption was associated with increased risk of overall cancer (HR = 1.12, 95% CI 1.03–1.23)
McCullough et al. (2022) [66]	934,777 participants	Prospective cohort study	Custom assessment of SSBs and ASB	Association between SSB and ASB and risk of cancer mortality	>2 SSB linked to obesity-related cancers (ICD C00-C97) (HR = 1.05; 95% CI 1.01–1.08); colorectal (HR = 1.09; 95% CI, 1.02–1.17;), and kidney (HR = 1.17; 95% CI, 1.03–1.34); ASB was not related to cancer. Pancreatic cancer was linked to ASB (HR = 1.11; 95% CI, 1.02–1.20)

* Outcomes reported are from the final regression models that controlled for body mass index (except where indicted), as well as other factors associated with developing cancers. ASB = artificially sweetened beverages; BMI = body mass index; CI = confidence interval; HR = hazard ratio; mL/d = milliliters per day; OR = odds ratio; SSB = sugar-sweetened beverages.

### 3.2. Preclinical Animal Studies

Preclinical studies have examined the effects of sucrose consumption on cancer outcomes and purported biological pathways driving disease processes. A study exploring the effects of high-sucrose diets compared with a starch diet in an APC^Min^ mouse model that spontaneously develops adenomas in the small intestine and colon showed a significant increase in the prevalence of colonic papillary tumors (32 of 54 mice vs. 19 of 63 mice) [18]. High-sucrose diets also increased the number of tumors in the proximal intestine (21.9 ± 1.4) compared to the control group (13.1 ± 1.6) [18].

Similarly, a study examining the effects of glucose and sucrose diets on hepatocarcinogenesis in rats that were exposed to the carcinogen diethylnitrosamine prior to being placed on the high sucrose diet found that the sucrose diet resulted in significantly heavier livers and two-fold more gamma-glutamyltranspeptidase-positive foci in the liver [67]. Another study using diethylnitrosamine to induce hepatocellular carcinoma found that mice receiving the carcinogen at 2 weeks of age and then fed high-sugar diets starting at 6 weeks of age through 32 weeks had significantly higher liver tumor burden, and numbers of tumors were significantly higher in mice fed high-sugar diets than in mice fed low-sugar diets irrespective of fat content [16]. Mice who consumed high-sugar diets had low adiposity but had significantly higher tumor burden compared to mice with high adiposity who consumed high-fat diets [16]. Importantly, overall body weights were not significantly different between groups [16]. The lack of an association between adiposity and liver tumor burden calls into question the theory that sugar increases cancer incidence via increased obesity [16,43].

The faulty reasoning linking sucrose with cancer only through obesity is corroborated by research from our laboratory [17]. Three breast cancer mouse models were used, with mice given an isocaloric non-sugar starch control diet or diets enriched with sucrose, fructose, or fructose plus glucose. Overall, the mice on the sucrose, fructose, and fructose-plus-glucose diets provided after tumor cell inoculation all exhibited significantly more widespread metastases to the lungs compared with those on the non-sugar control diets in the mice bearing mouse mammary carcinoma 4T1 orthotopic model [17]. Mice fed a high sucrose diet starting one day after injection of MDA-MB-231 cells also had increased tumor growth in the human breast cancer mouse orthotopic model. The third model used MMTV/neu mice that spontaneously develop mammary tumors and found that the mice fed the sucrose diet also developed significantly larger tumors and more rapid onset of breast cancer than did the control group [17]. Some argue that sugar’s harmful effects are due only to the unrealistically high amounts of sugar used in these types of preclinical studies. However, the sucrose-enriched diet used in our study was similar to the average sugar intake consumed in a Western diet. Furthermore, there was no statistically significant weight gain or difference in weights between any of the diet groups in all three tumor models, suggesting that sucrose, and especially fructose, plays an independent role in breast cancer risk and progression independent from body weight gain [17]. 

Although studies have indicated that glycemic load and its effects on the insulin pathway serve as the primary link between sugar and cancer, there are other potential mechanisms (Figure 2) [18,68,69,70,71,72]. Chronic inflammatory states with overexpression of cyclooxygenase or lipoxygenases have been associated with numerous cancer types and chronic diseases. More specifically, studies have found a strong association between 12-lipoxygenase (12-LOX) and its metabolites, 12-hydroxyeicostatetraenoic acid (12-HETE), and a variety of cancers [17,73,74]. Our study found that 12-HETE levels in breast tumors of mice fed sucrose-, fructose-, and fructose-plus-glucose-enriched diets were all significantly higher than those in mice fed a starch control diet [17]. This spike in 12-LOX/12-HETE levels due to the sugar-enriched diets suggests that inflammation, independent of weight gain or metabolism, is a novel causal mechanism in the association between sugar and cancer. Finally, a four-arm randomized controlled trial examining SSB consumption reported that fructose and sucrose (median basal hepatic fractional secretion rates (FSR)%/day: fructose 19.7 (*p* = 0.013); sucrose 20.8 (*p* = 0.0015); control 9.1) but not glucose increased liver lipid production, creating conditions for future adverse health outcomes [75].

A 2019 study published in *Science* demonstrated that HFCS enhanced intestinal tumor growth independently of weight gain [76]. When transgenic mice with APC deletion (APC^−/−^) were given 400 μL of 25% HFCS solution via oral gavage, providing calories from HFCS similar to human consumption of less than 355 g of SSB, for 8 weeks, the number of large adenomas (>3 mm in diameter) and high-grade tumors significantly increased in the HFCS group compared to a control group. Interestingly, chronic exposure to modest amounts of HFCS did not lead to obesity or metabolic dysfunction in the APC^−/−^ mice. This study also found that HFCS accelerated glycolysis by upregulating ketohexokinase and increasing de novo activation of the lipogenic pathway. Interestingly, the levels of 12-HETE in colon tumor tissues of HFCS-treated APC^−/−^ were significantly elevated compared with those of control mice, consistent with the results from Jiang et al. [17].

Other preclinical research published in *Cell* supports specific carbohydrate metabolic pathways linking fructose and liver metastases [77]. A study examining the effects of fructose on colon cancer liver metastases found that aldolase B (ALDOB), an enzyme that is involved in fructose metabolism, was upregulated in liver metastases compared with a normal colon and a primary colorectal cancer tumor [77]. Mice inoculated with human colorectal cancer cells and subsequently fed high-fructose diets had consistently increased liver metastases and shortened survival compared with control mice and with mice fed low-fructose diets [77]. In addition, mice with ALDOB knockdown had increased survival compared with the control group, suggesting that ALDOB is a potential target for fructose-induced liver metastases [77]. These are provocative findings given that cancer metastasis is the most common cause of cancer-related death and that liver metastases are common for almost every type of cancer. A greater understanding of the potential dangers of dietary fructose and sucrose regarding risk of cancer overall is critically important.

### 3.3. Human and Primate Studies of Sugar and Metabolic Syndrome

To experimentally test the effects of sucrose and fructose on the risk of cancer in humans, we would need to manipulate diets. Because cancer is complex and usually would take decades to manifest, human subjects research in this field is currently non-existent and is of ethical concern. However, compelling human and primate studies have explored the link between added sugar and metabolic syndrome (MetS). MetS is a cluster of medical risk factors that include high triglycerides, low high-density lipoprotein, high blood pressure, high fasting glucose, and central obesity. MetS is diagnosed when a patient has three of the five factors [78]. MetS is associated with an increased risk for cardiovascular diseases, diabetes, cognitive disorders, and other health conditions [79], including an increased risk for a number of common cancers, including breast, liver, pancreatic, colorectal, endometrial, and more [80]. The connections between MetS and cancer or between MetS and added sugar do not necessarily translate to a connection between cancer and added sugar. Nevertheless, substantial evidence suggests a causal link between MetS and added sugar [81], indicating important implications for our review.

In a rhesus monkey model, researchers found that 100% of monkeys fed a high-fructose diet had insulin resistance and other features of MetS [19]. Within 6–12 months, the high-fructose diet in monkeys produced central obesity, insulin resistance, inflammation (increased serum levels of C-reactive protein and monocyte chemoattractant protein-1), and dyslipidemia [19]. These results suggest that this rhesus monkey model of diet-induced obesity, insulin resistance, and dyslipidemia is directly translatable to MetS in humans [19].

Schwarz et al. explored the effects of fructose restriction in obese children [44]. Forty-one children aged 8–18 years with obesity and MetS whose normal diets consisted of large amounts of added sugars (fructose intake >50 g/day) were provided sugar-restrictive meals for 9 days that swapped sucrose and fructose for a calorically neutral and macronutrient-equivalent amount of starch [44]. Over that period, liver fat decreased from 7.2% to 3.8% [44]. In addition, fractional de novo lipogenesis decreased significantly in 37 of 40 participants (68% to 26%), including in those who did not lose weight, and insulin sensitivity increased significantly [44]. These results suggest that a reduction in sucrose was responsible for significantly lowered liver fat, visceral fat, and fractional de novo lipogenesis independent of weight loss [44]. Another study by Schwarz et al. used a similar design but with eight healthy men. This nine-day study explored the effects of a high-fructose diet compared with an equivalent macronutrient breakdown, with complex carbohydrates replacing the fructose [82]. Even though all the subjects maintained weight stability, those who consumed a high-fructose diet had significantly higher de novo lipogenesis and liver fat [82].

Other research has examined fructose-sweetened beverage versus glucose-sweetened beverage consumption in overweight or obese adult men and women and found that those who consumed fructose-sweetened beverages had significantly increased de novo lipogenesis, higher accumulation of intra-abdominal fat, a more atherogenic lipid profile, and reduced insulin sensitivity [83]. A follow-up study found that those who consumed the fructose-sweetened beverages had significantly decreased net postprandial fat oxidation and significantly increased net postprandial carbohydrate oxidation [84]. In addition, resting energy expenditure significantly decreased compared with baseline values in the fructose-consuming group [84].

The consistency in results across both primates and humans (children and adults) shows strong evidence of a direct causal link between fructose and MetS. Emerging data suggest a strong association between MetS and cancer risk [80,85,86,87,88], progression of disease [89], and mortality [85,89], although more research is needed to better understand the mechanisms. While the links between MetS, cancer, and added sugar remain unclear, the evidence connecting them is strong enough to warrant further research.

## 4. Discussion

The current review revealed evidence linking added sugar consumption to increased cancer incidence and mortality. The epidemiologic evidence was strongest for breast cancer [49,51,53], and we also identified studies examining and finding a connection between added sugar and colon cancer [11,55]. Research on the association between added sugar consumption and pancreatic cancer was mixed [12,13,57,58,59], yet the preponderance of the evidence suggests an association. Although some of the observational studies were prospective with large sample sizes [14,57,64,65], others had less robust designs with smaller samples [15,62,75]. Overall, the majority of the studies found an association between excess sugar consumption and cancer.

A critical question is whether the link between sugar and cancer is solely mediated by weight gain and obesity. Population-based studies on added sugar, especially SSB, and cancer risk and outcomes are equivocal on whether the association is driven by obesity or is also independent of obesity and weight gain. Some studies implicate a role for obesity [11,12,13,16], others show the enhanced risks independent of BMI and other lifestyle factors [44,90], and some suggest that the association may be cancer-specific [66,91]. However, as is the case with all observational studies, association does not mean causation, and further mechanistic and human clinical trials are needed.

In contrast, most preclinical research demonstrates that the effects of excess added sugar on cancer development and progression are independent of body weight gain [17,77]. Extensive research now supports the role for multiple mechanisms whereby sugar modifies cancer risk, independent of obesity, including inflammation, glucose/fructose metabolism, lipid metabolic pathways, and immune modulation (Figure 2) [17,76,77,92]. This suggests that obesity may have more of a bystander effect. Our review of the preclinical research revealed that high-sucrose or high-fructose diets activate several mechanistic pathways, including inflammation, glucose, and lipid metabolic pathways. Although human prospective studies linking sugar and cancer are limited, compelling human and primate studies have explored the link between added sugar and metabolic syndrome (MetS), a risk factor for cancer. Substantial evidence suggests a causal link between MetS and added sugar [19,44], supporting the association between excess sugar consumption and cancer. Therefore, it is the increased underlying inflammatory processes or alteration of metabolic pathways that may be driving the sugar–cancer link. Given the importance of inflammation in driving sugar-induced tumorigenesis and progression, it is logical for future research to investigate the role of the immune system in these processes. Overall, exploring the association between added sugar and cancer, in addition to other dietary constituents and patterns, independent of obesity, should be prioritized [43].

Perhaps the largest knowledge-gap comes from the lack of clinical trials research on humans. Such studies are not only near non-existent but also ethically challenging and would face time restrictions, as well as limited funding. Although animal studies show links between sugar and cancer that are independent of obesity [17,18], these models are not always translatable to humans. Both human and animal studies are needed to clarify sugar’s role in cancer and further explore the mechanisms of such effects. In the meantime, perhaps more caution is needed in how our population, and cancer patients in particular, are counseled in this area.

The current nutritional guidelines for cancer prevention and people with cancer remain silent on the harms of sugar and fructose consumption outside the context of weight gain, and perhaps a more precautionary message is needed [4]. Normal weight individuals may inappropriately believe the harms of sugar do not apply to them. As added sugar intake is increasing globally and added sugar consumption in the US far exceeds the ACS, AHA, and WHO recommendations for maximum intake, there is cause for concern that this modifiable risk factor is not being adequately addressed. Outside the context of cancer, excess sugar intake is linked with diabetes [93,94], cardiovascular disease [95], and Alzheimer’s disease and other forms of dementia [96] and is linked with other cause-specific deaths [97], and these associations are independent of obesity [93,94,95,97]. The underlying mechanisms are likely similar to those of cancer risk.

Without appropriate guidelines and regulatory changes, the general population will continue to experience sugar-induced health problems, including preventable cancers. As there is no research showing the *benefits* of consuming any amount of added sugar, and, given that added sugar is devoid of nutritional value, the recommended daily guidelines must reflect the health risks of sugar consumption independent of weight gain. It is also important that, as a society, we start to actually follow the guidelines established by the AHA, the ACS, and the WHO. This includes ensuring that food manufactures also reduce added sugars in their products. By using a system-wide approach to lowering sugar consumption, millions of premature deaths could be averted annually [98]. The general population and cancer survivors are entitled to and deserve appropriate counseling based on this evidence.

There are several limitations with the current review. The study did not set out to be a formal systematic review, and, as such, no specific search criteria were used to select research examining the link between sugar and cancer. Because of the dearth of research in the field of sugar consumption and cancer, we did not use specific selection criteria when choosing research studies to include. However, we tried to locate and highlight the most relevant and well-designed research that has been published to date. While we have cited epidemiologic, preclinical, and clinical studies that show a potential link between added sugar and cancer and those that do not, as well as explored plausible mechanisms, we understand that these findings are far from definitive.

## 5. Conclusions

In conclusion, research suggests a direct link between sugar and cancer. Preclinical studies and studies of people with MetS show that high-sucrose or high-fructose diets activate several mechanistic pathways, including inflammation, glucose, and lipid metabolic pathways, suggesting a causal link between excess sugar consumption and cancer development and progression that is independent of weight gain. Dietary guidelines and US policy need to reflect this new knowledge. Concerted action is needed to lower sugar intake in the US and other countries, better inform the public of the risks of excess sugar intake, and conduct more robust research in the field of added sugar and cancer.

## Figures and Tables

**Figure 1 cancers-14-06042-f001:**
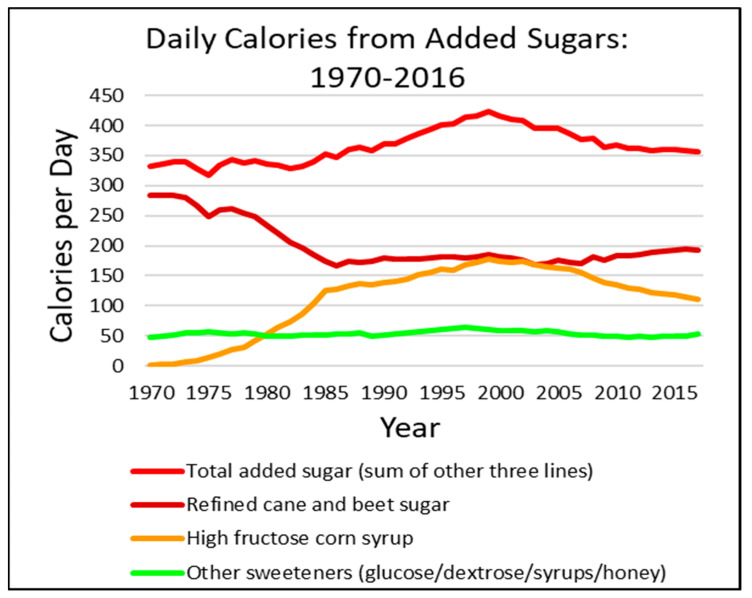
Average daily per capita calories from added sugars in the United States from 1970 through 2016. Food Availability (Per Capita) Data System; Loss-Adjusted Food Availability [5,8].

**Figure 2 cancers-14-06042-f002:**
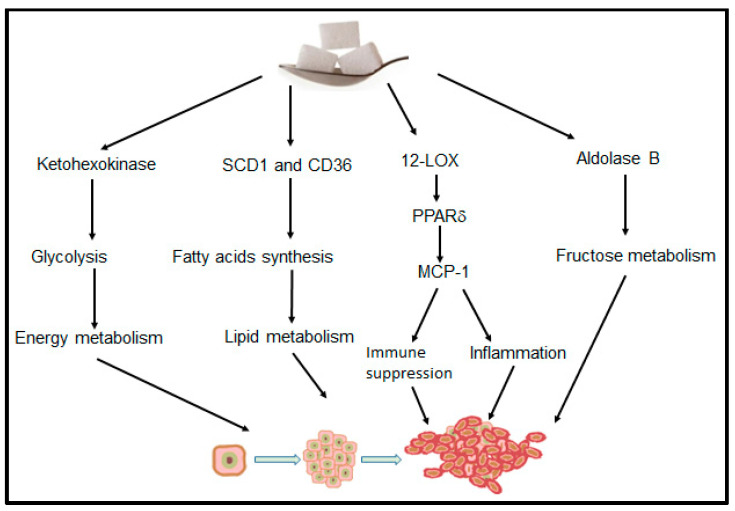
Proposed model whereby dietary sugar influences multiple cancer-specific pathways, including energy metabolism, lipid metabolism, inflammation, and immune function.

**Table 3 cancers-14-06042-t003:** Added sugar intake and risk of developing pancreatic cancer and mortality.

Author	Study Population	Study Design	Measure	Associations Examined	Main Findings *
Michaud, et al. (2002) [12]	88,802 women with pancreatic cancer	Prospective cohort study	Food frequency questionnaire	Association between carbohydrates, fructose, glycemic index, glycemic load, and sucrose and risk of pancreatic cancer	High glycemic load/fructose intake non-significantly associated with increase in pancreatic cancer risk (RR = 1.53, 95% CI 0.96–2.45). Among women who were both sedentary and overweight, high glycemic load (RR = 2.67, 95% CI 1.02–6.99) and high fructose intake (RR = 3.17, 95% CI 1.13–8.91) were significantly associated with increased pancreatic cancer risk
Larsson et al. (2006) [57]	77,797 adults with pancreatic cancer	Prospective cohort study	Food frequency questionnaire	Association between added sugar and high-sugar foods and pancreatic cancer	Highest vs. lowest category was significantly associated with pancreatic cancer for soft drinks (HR = 1.93, 95% CI 1.18–3.14); both added sugar (HR = 1.69, 95% CI 0.99–2.89), and sweetened fruit soups or stewed fruit (HR = 1.51, 95% CI 0.97–2.36) were not significantly associated with pancreatic cancer
Nothlings et al. (2007) [13]	162,150 adults with pancreatic cancer	Prospective cohort study	Food frequency questionnaire	Association between glycemic load, added sugar, carbohydrates, fructose, sucrose, and total sugars and pancreatic cancer risk	The highest vs. lowest quartile for fructose (RR = 1.35, 95% CI 1.02–1.80), fruit and juices (RR = 1.37, 95% CI 1.02–1.84), and sucrose in overweight/obese but not normal-weight participants (RR = 1.46, 95% CI 0.95–2.25) were associated with a greater RR of pancreatic cancer
Bao et al. (2008) [58]	487,922 adults with pancreatic cancer	Prospective cohort study	124-item food frequency questionnaire	Association between total added sugar and SSB intake and pancreatic cancer	No increased risk for pancreatic cancer was identified when comparing the highest quintile to the lowest quintile of added sugar intake (RR = 0.85, 95% CI 0.68–1.06)
Aune et al. (2012) [56]	10 studies with pancreatic cancer	Systematic review and meta-analysis of prospective cohort, or nested-case-control, or case-control studies	N/A	Association between carbohydrates, fructose, and glycemic indices and risk of pancreatic cancer	Positive association between fructose and pancreatic cancer risk (RR = 1.22, 95% CI 1.08–1.37) at 25 g/day. No other associations reached significance
Navarrete-Munoz et al. (2016) [59]	477,199 adults with pancreatic cancer	Prospective cohort study	Country-specific validated dietary questionnaire	Association between SBB intake and pancreatic cancer risk	Greater SBB intake was not associated with higher pancreatic cancer risk (HR = 1.03, 95% CI 0.99–1.07)

* Outcomes reported are from the final regression models that controlled for body mass index, as well as other factors associated with pancreatic cancer. ASB = artificially sweetened beverage; CI = confidence interval; HR = hazard ratio; RR = relative risk; SSB = sugar-sweetened beverages.

## Data Availability

Not applicable.

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
