# Peer review of "Understanding the Link between Sugar and Cancer: An Examination of the Preclinical and Clinical Evidence"

_cancers, 2022, doi:10.3390/cancers14246042_

Round 1

Reviewer 1 Report

The present article entitled, "Understanding the Link Between Sugar and Cancer: An Examination of the Preclinical and Clinical Evidence" written by Epner et. al is a well organized, relevant  and comprehensively described article. It focuses on linking Sugar and Cancer by examining the Preclinical and Clinical Evidences. This was a good, simple and an enjoyable read. 

Author Response

We are very grateful for the comments about our manuscript. 

Reviewer 2 Report

This is a nicely written review presenting the findings regarding sugar and cancer. The authors have corroborated the findings with dietary recommendations, making the article very interesting to read. The use of simple and clear language will broaden the reader base. To improve the article, the following are some recommendations:

1. Can you add a paragraph explaining Metabolic Syndrome?

2. In sub-section 1.1.2, India has been excluded although being the number one country for diabetes and ever-increasing cases of cancer. It is strongly recommended to discuss those findings.

3. Wherever animal studies involving tumors have been discussed, please specify if the tumors were implanted or induced.

Author Response

We are very grateful for the suggestions and comments about our manuscript. The manuscript has been revised by making relevant changes according to the comments from the reviewer.  A detailed response to each point is enclosed.  

  1. Can you add a paragraph explaining Metabolic Syndrome?

Response: We have added a definition and the potential risk of Metabolic Syndrome in Sub-section 3.3. on Page 16. Below is the new text:

“MetS is a cluster of medical risk factors that include high triglycerides, low high-density lipoprotein, high blood pressure, high fasting glucose, and central obesity. MetS is diagnosed when a patient has three of the five factors [78]. MetS is associated with an increased risk for cardiovascular diseases, diabetes, cognitive disorders, and other health conditions [79], including an increased risk for a number of common cancers including breast, liver, pancreatic, colorectal, endometrial, and more [80].”

  1. In sub-section 1.1.2, India has been excluded although being the number one country for diabetes and ever-increasing cases of cancer. It is strongly recommended to discuss those findings

Response: We agree with the reviewer. We have discussed the increased cancer incidence in India and overall increases of cancer incidence in the low- and middle-income counties, such as India and China in the sub-section 1.1.2 on Page 4.  Below is the new text:

“In India, cancer incidence increased 1.1 to 2.0 percent per year between 2010 and 2019 [28], with breast cancer being the most common cancer in women and lung cancer for men [24]. India and China also have the highest incidence and number of people living with diabetes [29], a known risk factor for many cancers [30]. By 2040, global cancer cases will increase by over 40% and it is estimated that two-thirds will occur in LMICs [31]. Changing diets including consumption of fast-foods, highly processed foods, and excess sugar consumption are hypothesized as a causative factor in the increasing incidence of cancer in LMICs.”

  1. Wherever animal studies involving tumors have been discussed, please specify if the tumors were implanted or induced.

Response: Thank you for the suggestion. We added the information about whether the tumors were implanted or induced along with the timing of the high sugar diet in the sub-section 3.2 Preclinical Animal studies on Page 3.